# Estimating Muscle Power of the Lower Limbs through the 5-Sit-to-Stand Test: A Comparison of Field vs. Laboratory Method

Luca Ferrari [1,2], Gianluca Bochicchio [1], Alberto Bottari [1], Francesco Lucertini [2], Alessandra Scarton [1,3] and Silvia Pogliaghi [1,4,*]

1 Department of Neurosciences, Biomedicine and Movement Sciences, University of Verona, 37131 Verona, Italy
2 Department of Biomolecular Sciences, University of Urbino, 61029 Urbino, Italy
3 Microgate Srl, 39100 Bolzano, Italy
4 Research Associate Canadian Center for Activity and Ageing, University of Western Ontario, London, ON N6A 3K7, Canada
* Correspondence: silvia.pogliaghi@univr.it; Tel.: +39-045-8425128

**Abstract:** The 5-Sit-to-stand test (5STS) is used for lower limb muscle power (MP) determination in field/clinical setting. From the time taken to perform five standing movements and three partially verified assumptions (vertical displacement, mean concentric time, and mean force), MP is estimated as the body's vertical velocity x force. By comparison with a gold standard, laboratory approach (motion capture system and force plate), we aimed to: (1) verify the assumptions; (2) assess the accuracy of the field-estimated MP ($MP_{field}$); (3) develop and validate an optimized estimation ($MP_{field-opt}$). In 63 older adults (67 ± 6 years), we compared: (i) estimated and measured assumptions (2-WAY RM ANOVA), (ii) $MP_{field}$ and $MP_{field-opt}$ with the reference/laboratory method ($MP_{lab}$) (2-WAY RM ANOVA, Pearson's correlation coefficient (r), Bland-Altman analysis). There was a significant difference between estimated and measured assumptions ($p < 0.001$). Following the implementation of the optimized assumptions, $MP_{field-opt}$ (205.1 ± 55.3 W) was not significantly different from $M_{lab}$ (199.5 ± 57.9 W), with a high correlation (r = 0.86, $p < 0.001$) and a non-significant bias (5.64 W, $p = 0.537$). Provided that corrected assumptions are used, 5STS field test is confirmed a valid time- and cost-effective field method for the monitoring of lower limbs MP, a valuable index of health status in aging.

**Keywords:** sit-to-stand test; muscle power; aging

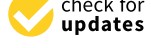

## 1. Introduction

Muscle strength is a key determinant of current and prospective health, mobility, and independent living in aging [1,2]. The loss in muscle function (i.e., strength, power, and endurance) is associated with increased adverse outcomes, including cardiometabolic diseases, falls, functional decline, frailty, and mortality [3]. Among the components of muscle function, muscle power, defined as the product of the force or torque of a muscular contraction and its velocity [4], seems to be the strongest predictor of functional limitations in aging [5–7]. Indeed, the tasks of real life require some amount of muscle power to guarantee independent living and to overcome gravity [8]. Moreover, muscle power declines with a faster dynamic [9] compared to strength, making it an ideal sentinel index for detecting individuals at higher risks and monitoring aging trajectory [10].

The gold standard quantitative analysis of muscle power is typically performed in a laboratory setting with the use of instruments such as the Motion Capture System (Mo-Cap), force platform, or isokinetic dynamometer [6]. However, the high costs and the limited accessibility of such equipment, the need for highly specialized personnel, and the complexity of the set-up make these measures not applicable on a large scale and/or in clinical settings [6,11]. To overcome these limitations, some field methods have been developed to evaluate muscle power in a time- and cost-efficient way [5]. Among them,

the 5-Sit-To-Stand test (5STS) has been preferred over other methods (e.g., vertical leaps, long jumps, stair climb, force-velocity profiling) [12–15] for testing elderly populations in clinical settings thanks to its low-risk profile and relationship with functionally relevant movements [2,14].This test requires five sit-to-stand movements to be performed from a box of a standardized height as fast as possible. The estimation of mean concentric lower limb muscle power is performed using the formula proposed by Alcazar et al. [6]. This formula relies on basic physics principles and on the following assumptions: (i) the duration of the concentric phase is approximated as one-tenth of the total time trial; (ii) the vertical displacement is calculated as the difference between the lower limb length (assumed to be equal to 50% of the subject's stature) and the height of the box; (iii) the force expressed during the concentric phase of standing is equal to 90% of the subject's body mass multiplied by g (9.81 m $\times$ s$^{-1}$) (for the formula and a more detailed description of the parameters, please see the Section 2.5).

This approach has increasingly been used in clinical settings [1]. However, the assumptions on which it is based have only been partially scrutinized [16]; therefore, a verification against a fully objective, gold standard approach is lacking. Finally, the replication of the validation process of the lower limb muscle power estimates outside the laboratory settings in which the method was developed is warranted.

By comparison with a fully automated gold standard method (motion capture system and force platform), the aim of this study was to: (1) verify the accuracy of the assumptions of Alcazar's method (Duration, Position, and Force); (2) verify the accuracy of the estimates of muscle power with Alcazar method; (3) develop and validate an optimized estimation of muscle power that are based on optimized assumptions.

## 2. Materials and Methods

### 2.1. Subjects

Sixty-three older adults were recruited (thirty-four males, $68 \pm 6$ years and twenty-nine females, $66 \pm 6$ years).

Participants eligible for this study were at least 60 years old of age and free of any orthopedic, mental, or neurological disease that could have interfered with the ability to express lower limb maximal power, see Table 1. All participants signed a written informed consent prior to participation. All procedures used were approved by the Ethics Committee for Human Research from the University of Verona and conducted in conformity with the Declaration of Helsinki (28/2019).

**Table 1.** Anthropometric characteristics of the participants (*N* = 63).

| | Age (Years) | | Body Mass (kg) | | Stature (m) | | BMI | |
|---|---|---|---|---|---|---|---|---|
| | **F** | **M** | **F** | **M** | **F** | **M** | **F** | **M** |
| *N* | 29 | 34 | 29 | 34 | 29 | 34 | 29 | 34 |
| Mean $\pm$ SD | $66.0 \pm 5.5$ | $68.2 \pm 6.3$ | $65.9 \pm 11.7$ | $80.4 \pm 14.4$ | $1.610 \pm 0.061$ | $1.743 \pm 0.633$ | $25.4 \pm 4.3$ | $26.6 \pm 5.0$ |
| Range (min-max) | (60.0–79.0) | (60.0–81.0) | (48.9–104.7) | (54.8–124.7) | (152.3–174.7) | (161.7–184.2) | (20.3–37.4) | (17.5–39.9) |

SD, standard deviation; F, females; M, males; BMI, Body Mass Index.

### 2.2. Procedures

All participants visited the laboratory once. During the visit, participants performed the 5STS test for lower limb muscle power determination (see Section 2.4 for more details). Participants' anthropometric measures were also collected prior to testing (see Section 2.3 for more details).

### 2.3. Anthropometric Measures

The anthropometric assessment was performed with participants barefoot and wearing only underwear. Body mass was taken to the nearest 0.1 kg with an electronic scale (Tanita electronic scale BWB-800 MA, Tokyo, Japan) and stature was measured with a Harpenden

stadiometer (Holtain Ltd., Crymych, Pembs, UK) to the nearest 0.005 m. Body Mass Index (BMI) was calculated as body mass/height$^2$ (kg/m$^2$).

*2.4. 5STS Test*

All participants performed a 10-min warm-up protocol immediately before the test, consisting of 5-min cycling on a cycle-ergometer at a fixed power and cadence (i.e., 50 W at 60 rpm), 4 active mobility exercises for the upper and lower limbs, and 5–6 repetitions of the sit-to-stand movement, which was also considered as a familiarization to the 5STS test. A full description of the STS setup is detailed in Figure 1 caption.

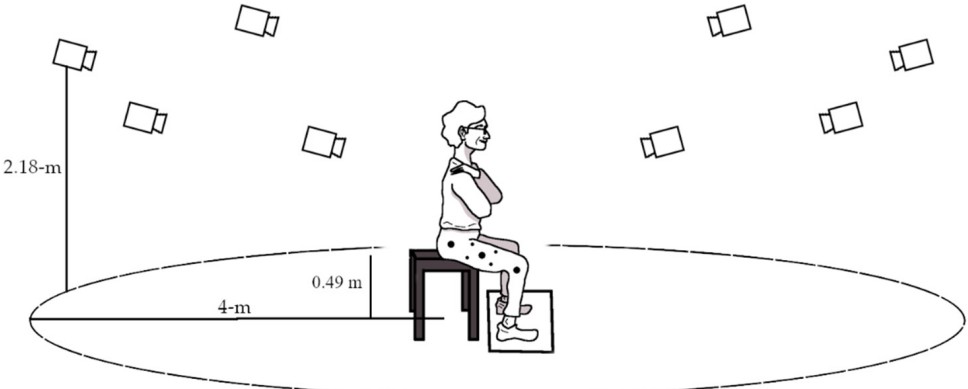

**Figure 1.** Schematic representation of the data collection setup. The figure represents the position of the lower limbs at the beginning of the test. The participant was sitting on the box with the trunk and shanks positioned perpendicular to the ground. The larger black dots represent the main markers placed on the trochanter, mid-thigh, and lateral epicondyle of the femur (from left to right), respectively. The marker set was completed by a total of 4 additional markers used for signal recovery purposes (smaller black dots) positioned around the mid-thigh main marker. The force platform is represented as the white square under the participant's feet. A total of 8 cameras were placed at 2.18-m height and 4-m distance around the participant.

Participants started the test from a sitting position on a 49-cm box, keeping the trunk and shanks perpendicular to the ground and the arms crossed on their chest. On the count of three, participants performed the 5STS test, consisting of 5 consecutive repetitions of the sit-to-stand movement, to be executed as fast as possible. Participants concluded the test in the same initial sitting position. The test was performed twice, with a 3-min recovery between the two trials [4,17,18]. To ensure safety and prevent any risk of falls, a researcher was close to the subject throughout the entire duration of the tests. A stopwatch was used to measure the total time of each trial. Kinetic and kinematic data were recorded with a Motion Capture system (MoCap) composed of 8 infrared cameras (Vicon, Oxford, UK) automatically synchronized with a force platform (AMTI Inc., Watertown, MA, USA) positioned under participants' feet. A Marker set of 7 markers was placed on participants' right lower limb (Figure 1) to track lower limb trajectory. The MoCap system and the force platform outputs were recorded at 200 Hz and 1000 Hz, respectively.

*2.5. STS Test Analysis*

2.5.1. Field Method

Lower limb muscle power of the two trials was first estimated with the formula proposed by Alcazar et al. [6], using variables and calculations reported in Table 2.

$$STS\ mean\ power = \frac{Body\ mass \times 0.9 \times g \times [Height \times 0.5 - Chair\ height]}{Five\ STS\ time \times 0.1}$$

**Table 2.** Variables of the Alcazar's calculation of sit-to-stand (STS) mean power.

| Variables (Unit of Measurements) | Formula | Notes |
|---|---|---|
| Vertical displacement (m) | (50% × stature) − box height | Stature in meters; box height = 0.49 m |
| Concentric time (s) | 10% × Total trial duration | Total trial duration in s |
| STS mean velocity (m/s) | Vertical displacement/Concentric time | |
| STS Force (N) | 90% × Body Mass × g | Body Mass in kg; g = Gravity acceleration 9.81 m/s$^2$ |
| STS mean Power (W) | STS Force × STS mean velocity | |

### 2.5.2. Laboratory Method

Participants' kinetic and kinematic variables were measured in both trials using the force platform and MoCap system (on *z*-axis, perpendicular to the ground). Trochanter vertical displacement and vertical velocity were automatically extrapolated by the system and used to make all the subsequent computations. Position tracking and force signals were low pass filtered at 7 and 15 Hz, respectively, using a Butterworth filter of second order.

The identification of the concentric and eccentric phases, as well as the total trial duration, is depicted in Figure 2 and required the following steps: (1) identification of all zero-crossing events. In brief, since the velocity is the derivative of the position, its mathematical sign changes during the STS movement in accordance with posture changes. Therefore, the intersection of the velocity signal with the "0-axes" was identified as a moment in which the subject was momentarily still in either a standing or sitting position; (2) establishing the correspondence between zero-crossing events, body position, and movement phase. After the identification of the zero-crossing events, their correspondence with the actual body position and movement phase for each repetition was established based on the following rules and numbered progressively:

- sitting position and start of the concentric phase (STARTC1-5) = the zero-crossing moment is followed by a positive peak in the velocity signal.
- standing position and stop of the concentric phase (STOPC1-5) = the zero-crossing moment is preceded by a positive peak in the velocity signal.
- re-sitting position and stop of the eccentric phase (STOPE1-5) = the zero-crossing moment is preceded by a negative peak in the velocity signal.

### 2.5.3. Calculations

- Time: the total duration of the 5STS Test was computed as the difference between the time coordinates of STARTC1 and STOPE5. The duration of the single concentric phases was calculated as the difference between the time coordinates of STARTC and STOPC (STS 1–5). Thereafter, the duration of the five single concentric phases was averaged and expressed as % of the total test duration. The durations of the eccentric phases were calculated as the difference between the time coordinates of STOPC and STOPE (STS 1–5).
- Displacement: the time coordinates of the above-defined significant events were also used for the processing of the position signal. (Figure 3, middle panel). The displacement of the concentric phases was calculated as the difference between the space coordinates of STARTC and STOPC (STS 1–5). The displacement of the eccentric phases was calculated as the difference between the space coordinates of STOPC and STOPE (STS 1–5).
- Velocity: for each STS repetition, the mean concentric velocity was computed as the ratio between displacement and duration of the concentric phase.
- Force: The mean concentric force was calculated as the average of the ground reaction force signal within each concentric phase (i.e., within the time coordinates of STARTC and STOPC 1–5) (Figure 3, bottom panel).

- Power: for each repetition, the lower limb muscle power was computed as the product between mean concentric velocity and mean concentric force.

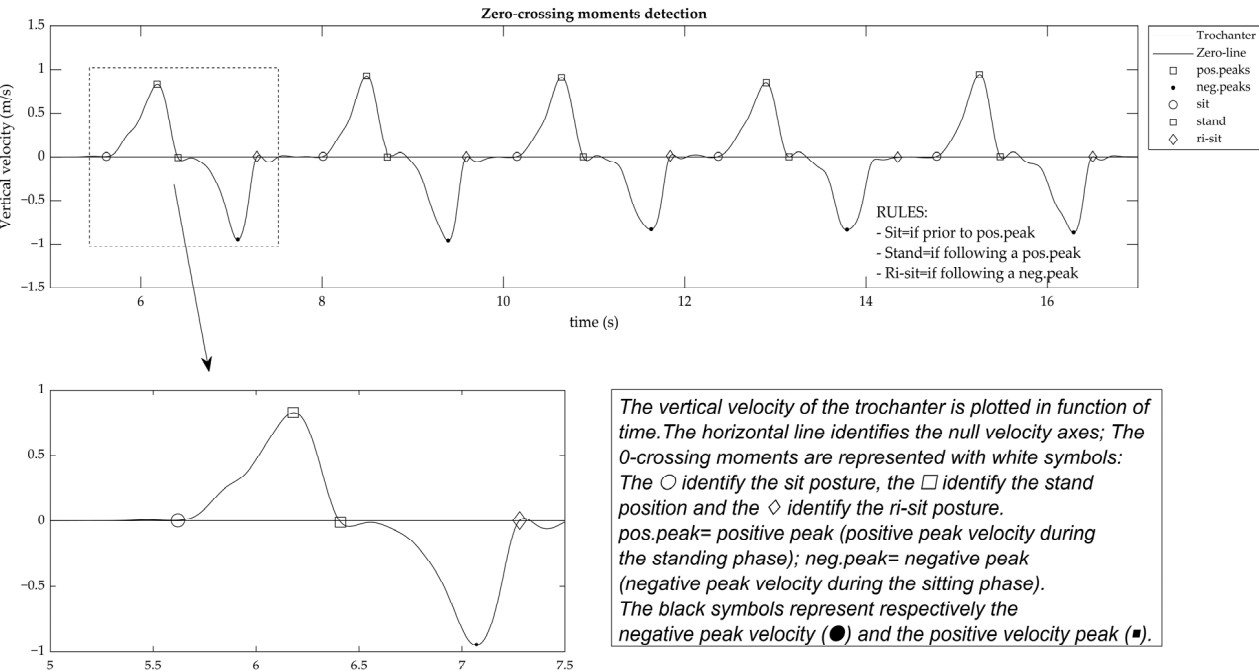

**Figure 2.** Example of vertical velocity profile during a 5STS Test as a function of time.

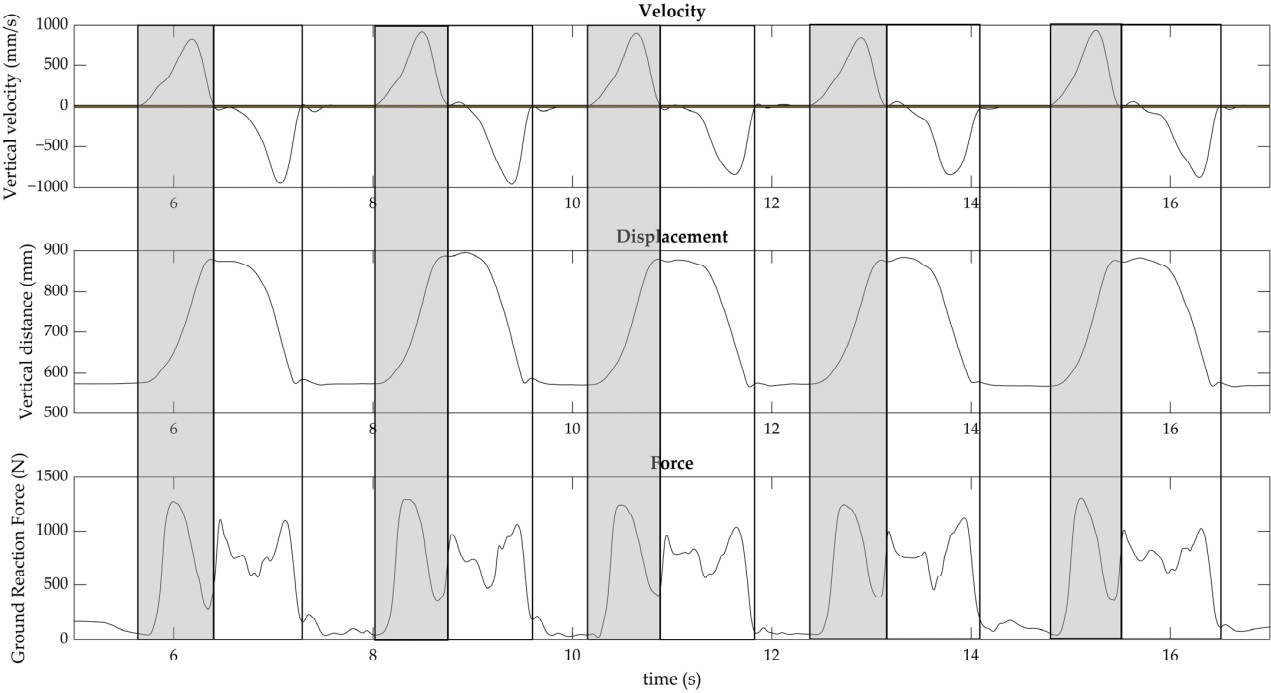

**Figure 3.** Sample data from a representative subject. Trochanter vertical velocity (the horizontal line identifies the null velocity axes), trochanter vertical distance (displacement), and ground reaction force on *z*-axes as a function of time. The grey rectangles identify the concentric portion of the movement (i.e., the standing phase). The white rectangles identify the eccentric portion of the movement (i.e., the sitting phase).

### 2.6. Optimised Field Method

Since the Alcazar formula was based on partially verified assumptions, we wanted to optimize it through the values measured with the laboratory approach and to test the validity of the optimized version of the field method. Therefore, the Alcazar's formula was implemented substituting the original assumptions with the group-average values of Time, Position, and Force values found with the laboratory approach (Table 3).

We decided to replace Alcazar's parameters with laboratory-based group averages for the following reasons: (i) the standard deviation of all measured parameters (duration (%total time), position (%chair height and stature), and force (%body weight)) was rather small (see Table 3). In particular, ~12.5% for duration, ~3% for position, and ~10% for the force, leaving little room for individualized predictions; (ii) all parameters were not correlated with either age, stature, or BMI and were not different among sexes. Moreover, an attempted multiple linear regression analysis with duration (i.e., the most variable parameter) as Y and sex, age, stature, and BMI as predictors yield a not significant result. Therefore, we were unable to find simple predictors that would allow an individualized estimate of the elements of Alcazar's calculations. (iii) Last but not least, our intention was to develop a ready-to-use and generalizable formula that would retain the simplicity of the original Alcazar's approach. In summary, a population average estimate of the parameters was considered a fair approach for parameters optimization.

To test the validity of this optimized formula, we recruited a subgroup of our participants based on the will to return in follow-up research (after 3 months). These 42 individuals (20 females, 22 males, age: $67.4 \pm 6.8$ years, stature: $1.6711 \pm 0.09$ m, body mass: $73.4 \pm 12.4$ kg, BMI: $25.3 \pm 4.0$ kg/m$^2$) performed a new trial of the 5STS test with the same procedure earlier described in the Section 2.4 of the methods. We used this as an independent sample for the validation process: the individual kinetic and kinematic variables were measured with the laboratory method (see Section 2.5.2) and with field method and optimized assumptions corresponding to the group average values of duration, position, and force (Table 3).

All the calculations were automatically performed by a MATLAB (MathWorks, Natick, MA, USA) script and exported on an Excel (Microsoft, Washington, DC, USA) spreadsheet for subsequent calculations. The values obtained from the five sit-to-stand repetitions within the test were averaged.

### 2.7. Statistical Analysis

All data were checked for normality using the Shapiro–Wilk test. As a preliminary evaluation, a two-way repeated-measures ANOVA (TRIAL × METHOD) was performed to test differences in the mean concentric time (expressed as % of the total time), mean vertical displacement (initial and final positions expressed as % of box height and participants' stature, respectively), and mean force values (expressed as % of body weight) between the two trials. Since there was no main significant effect of TRIAL on the outcomes, the values of the two trials were averaged for each participant. A two-way, repeated measures ANOVA (SEX × METHOD) was performed to test differences in the time, position, and force between the assumed Alcazar's and the laboratory-measured values.

**Table 3.** Elements of the Alcazar's calculation of sit-to-stand (STS) mean power.

| | | Males (n = 34) | | | Females (n = 29) | | | Total (n = 63) | | | Main Effect | | |
|---|---|---|---|---|---|---|---|---|---|---|---|---|---|
| | | Estimated | Measured | p | Estimated | Measured | p | Estimated | Measured | p | Method | Sex | Interaction |
| Duration | Concentric phase (% total time) | 10.0 ± 0.0 | 8.3 ± 1.0 | <0.001 | 10.0 ± 0.0 | 8.1 ± 1.0 | <0.001 | 10.0 ± 0.0 | 8.2 ± 1.0 | <0.001 | <0.001 | 0.55 | 0.55 |
| | Concentric time (s) | 1.0 ± 0.2 | 0.8 ± 0.1 | <0.001 | 1.0 ± 0.3 | 0.8 ± 0.1 | <0.001 | 1.0 ± 0.3 | 0.8 ± 0.1 | <0.001 | <0.001 | 0.962 | 0.302 |
| Position | (% chair height) | 100.0 ± 0.0 | 118.5 ± 4.3 | <0.001 | 100.0 ± 0.0 | 118.9 ± 3.2 | <0.001 | 100.0 ± 0.0 | 118.7 ± 3.8 | <0.001 | <0.001 | 0.801 | 0.801 |
| | Stop (% stature) | 50.0 ± 0.0 | 53.9 ± 0.0 | <0.001 | 50.0 ± 0.0 | 54.5 ± 0.0 | <0.001 | 50.0 ± 0.0 | 54.2 ± 2.0 | <0.001 | <0.001 | 0.148 | 0.148 |
| | Displacement (m) | 0.4 ± 0.03 | 0.3 ± 0.1 | <0.001 | 0.3 ± 0.0 | 0.3 ± 0.0 | <0.001 | 0.3 ± 0.0 | 0.3 ± 0.0 | <0.001 | <0.001 | <0.001 | 0.422 |
| Velocity | Mean concentric Velocity (m/s) | 0.4 ± 0.1 | 0.5 ± 0.1 | <0.001 | 0.3 ± 0.1 | 0.4 ± 0.1 | <0.001 | 0.3 ± 0.1 | 0.4 ± 0.1 | <0.001 | <0.001 | <0.001 | 0.765 |
| Force | % BW | 90.0 ± 0.0 | 69.2 ± 5.3 | <0.001 | 90.0 ± 0.0 | 67.7 ± 8.26 | <0.001 | 90.0 ± 0.0 | 68.5 ± 6.8 | <0.001 | <0.001 | 0.402 | 0.402 |
| | Newton | 710.2 ± 126.6 | 543.7 ± 92.0 | <0.001 | 583.7 ± 102.9 | 442.2 ± 120.7 | <0.001 | 583.7 ± 102.9 | 490.1 ± 99.9 | <0.001 | <0.001 | <0.001 | 0.062 |

A two-way, repeated measures ANOVA (SEX × METHOD) was used to test differences between field-estimates and laboratory measures of muscle power. Moreover, the absolute level of agreement between estimated and measured power was investigated using Pearson's correlation coefficient (r), Bland-Altman analysis. The correlation coefficient was interpreted according to the values of the r: trivial (<0.1); small (0.10–0.29); moderate (0.30–0.49); large (0.50–0.69); very large (0.70–0.89); extremely large (0.90–1.00) [19]. The Bland–Altman analysis [20] was followed by a one-sided z test on the bias.

The same approach (i.e., two-way, repeated measures ANOVA (SEX × METHOD), Pearson's correlation coefficient (r), Bland–Altman analysis) was finally used to compare new estimates of muscle power calculated by using optimized assumptions against the laboratory measures of muscle power.

Data are reported as mean ± SD, unless otherwise stated. Level of significance was set at 0.05. The SigmaPlot software (SigmaStat, San Jose, CA, USA) was used to conduct all the statistical analyses 12.0.

## 3. Results

Time, position, and force assumptions of male and female participants are reported in Table 3. A significant main effect of method was found for all the variables tested, but not for sex when expressed in relative terms (%). No significant SEX X METHOD interactions were found on any of the variables tested.

A significant main effect of methods is indicated with the *p*-value (<0.05). Significant differences between methods within each sex category are indicated with the *p*-value (<0.05); post hoc comparisons were not performed in the absence of a main effect of methods. BW = Body Weight.

There was a main effect of method ($p < 0.001$) and sex ($p < 0.001$) on mean muscle power (Table 3). Moreover, the Bland–Altman analysis showed a significant bias between measures with a relatively poor precision as reported in Table 4.

**Table 4.** Comparison between estimated and measured muscle power.

| | Groups | | | Main Effect | | |
|---|---|---|---|---|---|---|
| | **Males** | **Females** | **All** | **Method** | **Sex** | **Interaction** |
| MP lab (Watt) | 241.4 ± 51.1 | 171.0 ± 72.0 | 209.0 ± 70.6 | <0.001 | <0.001 | 0.227 |
| MP field (Watt) | 281.2 ± 66.6 | 195.7 ± 70.0 | 241.9 ± 80.1 | | | |
| Statistical Significance (*p* < 0.05) | <0.001 | 0.009 | <0.001 | | | |
| Correlation (r) | 0.58 | 0.84 | 0.79 | | | |
| Bias (mean differences) | 39.8 | 24.7 | 32.9 | | | |
| Z-score | 4.2 | 3.4 | 5.3 | | | |
| Imprecision (S.D.) | 55.6 | 39.8 | 49.2 | | | |

Mean and standard deviation of estimated and measured mean power in males, females, and overall group are presented along with the results of the statistical analysis: Two-way RM ANOVA (METHOD and SEX), Pearson product moment correlation, and Bland–Altman analysis (bias and precision).

Estimated muscle power values obtained with the modified assumptions (i.e., optimized field approach) in a separate data pool did not significantly differ from the measured ones. Furthermore, estimated and measured muscle power were highly and significantly correlated, with a relatively small standard error of estimate. Finally, the Bland–Altman analysis showed no significant bias between measures (Figure 4).

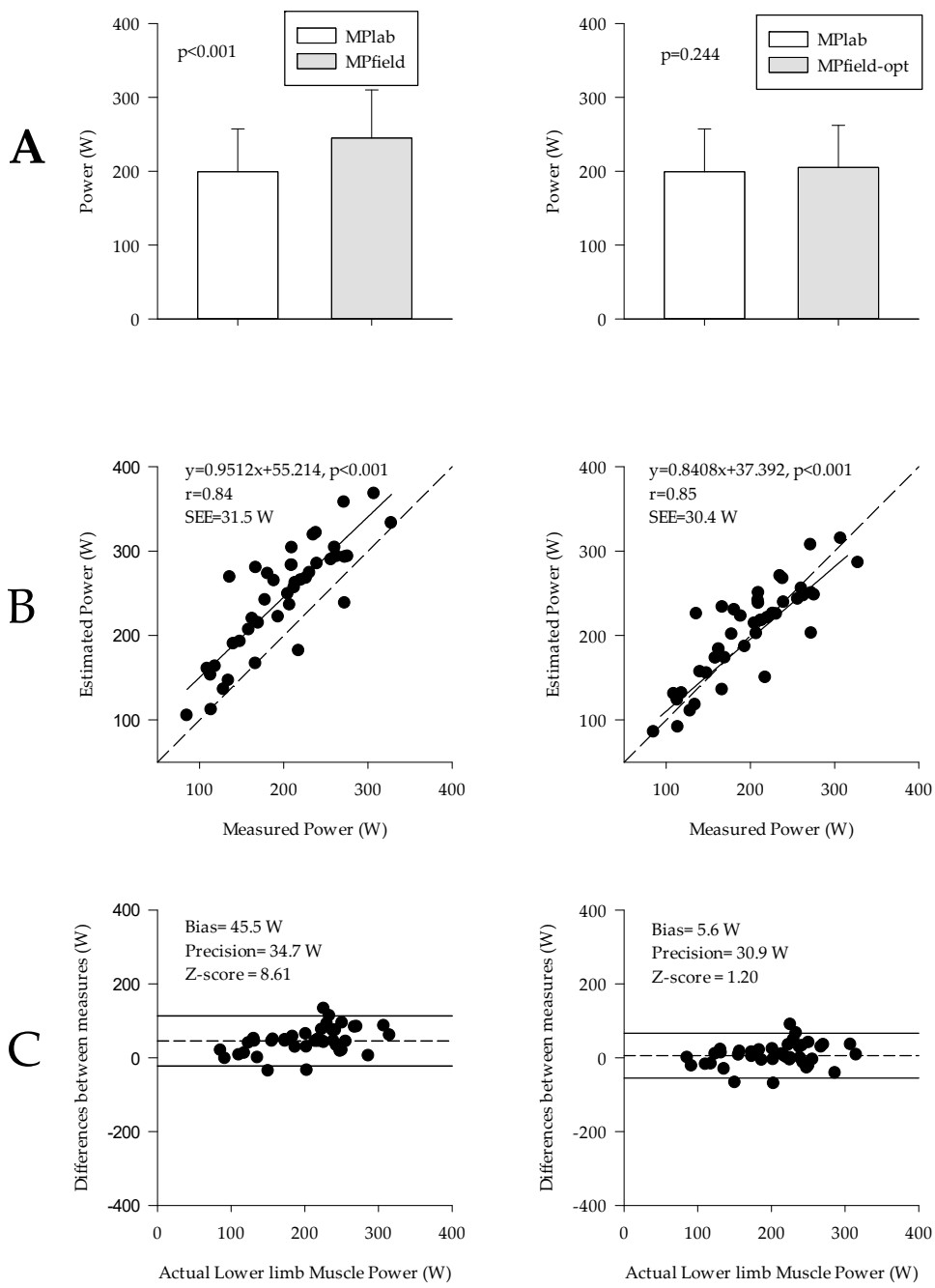

**Figure 4.** On the left side of the figure, we reported the comparison of means, correlation graph and Bland–Altman analysis referred to the Alcazar's method vs. Laboratory method. On the right side of the figure, we reported the same panels referred to the Alcazar's method with modified assumptions vs. Laboratory method. Panel (**A**): Comparison between estimated vs. measured mean values of lower limb muscle power. *p*-values are reported. Panel (**B**): Relationships between estimated vs. measured values of lower limb muscle power. Equation, *p*-value, Pearson's correlation coefficient (r), and SEE are reported along with regression (solid line) and identity (dashed line) lines. Panel (**C**): Individual differences between measured and estimated lower limb muscle power are plotted as a function of the mean of the two. Bias (i.e., the mean difference between measures; dashed lines) and precision (i.e., limits of agreement; solid lines) are displayed along with numerical values and the results of the one-tail z test on the bias.

## 4. Discussion

This is the first study to test the accuracy of the assumptions of Alcazar's method by using a fully objective, gold standard, and automated method. The main findings showed that the original Time, Position, and Force assumptions of the Alcazar's method were all overestimated when compared with the laboratory method. In addition, the estimation of the lower limb muscle power significantly overestimated the measures obtained with the laboratory method. Finally, when an optimized set of assumptions was implemented in the Alcazar's formula, the $MP_{field-opt}$ was proved valid, highly correlated, and with no significant bias compared to the $MP_{lab}$. Therefore, as long as validated assumptions are used, the 5STS field test is confirmed a valid tim- and cost-effective field method for the assessment of lower limbs MP.

### 4.1. Time Assumption

Regarding the time assumption, Alcazar's equation [6] estimates the mean concentric time as one-tenth or 10% of the total trial duration because it assumes that (i) no rest between repetitions is present, i.e., the total trial time is "active time" and that (ii) concentric and eccentric portions of a single repetition have an equal duration (50%) and [6,16]. On the contrary, in the current study, "active time", or time actually spent in concentric or eccentric actions, is in fact only 87% of the total time trial. In fact, about 13% of the time was actually occupied by a "transition phase" between repetitions. During this phase, participants take contact with the box and perform an oscillatory movement (backward and forward) of the torso before standing up again. Moreover, concentric and eccentric portions of a single repetitions were different from the estimated 50% (concentric portion = 47.7% $\pm$ 4.9; eccentric portion = 52.9% $\pm$ 5.1). The combination of the above factors may explain why the single mean concentric time (expressed as % of the total time) measured in our study was in fact was 8% and not 10%, as originally assumed by Alcazar's method.

The practical consequences of the overestimation of single mean concentric phase by Alcazar's method is an underestimation of the movement velocity.

### 4.2. Position Assumption

Alcazar et al. [6] calculated the vertical distance covered by the center of mass during the sit-to-stand movement as the difference between the lower limb length (assumed equal to 50% of the participants' stature) and the height of the box/chair. This assumption found a confirmation in a recent study that measured the lower limb length with a DXA scan, reporting it to be on average 51% of the participants' stature [16]. However, in the present study, we found a greater value for this parameter (55% of the participants' stature). This discrepancy may lie in the different instruments used (DXA-scan vs. MoCap system).

Moreover, in this study, we measured the actual height of the trochanter during the sitting position, whereas all the previous studies [6,11,14–16] considered the chair height as the bottom coordinate for the computation of the vertical displacement. We found a 19% difference (~9 cm, range 5–15 cm) between the measured and estimated height of the trochanter from the ground during sitting. Interestingly, we found a weak correlation between the distance of the trochanter from the chair seat and the individual's BMI. We may speculate that subjects who may be either fatter or with larger gluteal muscle masses may start their sitting movement from a higher position. In any case, the starting position of the trochanter is always different from the chair height. Again, the practical impact of incorrectly calculating the vertical displacement, as the difference between 50% of the participants' stature and the box height, leads to an overestimation of the distance parameter.

### 4.3. Force Assumption

Alcazar's method [6] assumed that the force expressed during the concentric phase of standing is equal to the 90% of the participants' body mass multiplied by g (9.81 m/s$^2$). The rationale behind the assumption is that (i) only the portion of the body that is above

the knee will be accelerated during the task and (ii) a minimum force equal to the body weight multiplied by gravitational force is required to accelerate the body.

Again, the above assumptions find a recent corroboration in Fernandez et al. [16], who used a force platform and found force values ~87% of body weight. In our study, however, we found force values that corresponded to only 67% of the participants' body weight. This may be due to the different methods applied to identify the concentric phase span in Fernandez study (subjective identification based on 2-D video) vs. our work (automatic identification based on the velocity profile MoCap system). It should be noted that body mass multiplied by g may in fact not be an adequate index of the force expressed by the subject. Indeed, this value represents a gravitational force rather than the actual force exerted over time by a subject during the standing phase [21]. On the contrary, from a mathematical perspective, force is the product between a body mass and its acceleration. During a standing movement, participants' body mass is constant and the variations in acceleration lead to a proportional variation in the force signal. During a standing movement, there is a "propulsive portion" (a rise in the force signal) followed by a "deceleration portion" (a drop in the force signal) in the proximity of the standing posture (Figure 3, bottom panel). The computation of the mean concentric force is directly influenced by these two portions and may in fact not be reflected by the simplistic mass multiplied by g calculation. In conclusion, using 90% of the participants' body mass multiplied by g ($9.81$ m/s$^2$) for the estimation of the force appears to be an incorrect overestimation.

### 4.4. Power Estimation

The sit-to-stand is an everyday life movement [22] that involves the engagement of the trunk muscles and lower limbs, postural control, and joints coordination [2]. These features make it a more relevant movement in terms of functionality over the analytic, guided, and single-joint movements (e.g., knee extension) and justify the use of 5STS Test as a commonly used field test for the estimation of lower limb muscle power [6,16]. Alcazar's formula has the merit of simplifying and facilitating the assessment of lower limb muscle power on a large scale and/or in clinical settings. A recent study determined that, although the Alcazar method significantly overestimated force and underestimated velocity, the resulting calculation led to an accurate estimate of the muscle power as measured with a partially automated laboratory approach [16]. Our current study confirmed an overestimated force and underestimated velocity in the Alcazar method vs. a fully automated and objective laboratory method. However, the discrepancy being large, our study identified a significant and not negligible (i.e., ~15%) overestimation of the field method vs. the reference, laboratory test (estimated muscle power: $241.9 \pm 80.1$ W; measured muscle power: $209.0 \pm 70.6$ W, $p < 0.001$).

The implementation of Alcazar's formulas with optimized assumption parameters, in a separate set of tests, led to lower limb muscle power that did not significantly differ between the two methods (optimized estimated muscle power: $205.1 \pm 55.3$ W; measured muscle power: $199.5 \pm 57.9$ W, $p = 0.537$). Furthermore, the optimized estimated lower limb muscle power was significantly and highly correlated with the muscle power measured using the gold standard methodology ($r = 0.85$) with a not significant bias between measures (MP: 5.6 W; z-score = 1.20).

### 4.5. Limitations and Future Developments

The present study has some limitations. While we included participants of both sexes and a relatively large age range (60–81 years), we excluded individuals who were affected by neurological, orthopedic, or mental diseases. Therefore, the conclusions of the study and the optimization parameters may not extend to younger/older population and/or to specific clinical populations. Another limitation is that we used a box of standardized height independently of the anthropometric characteristics of the individuals. This could lead to a different vertical distance to be covered from the sitting-to-standing position [23] and so to

different mechanical work and muscle energy expenditure. Therefore, a musculoskeletal model for estimating muscle energy could be considered for future studies.

## 5. Conclusions

The 5STS is a valuable and common test used to estimate lower limb muscle power thanks to its simplicity and low cost. However, the assumptions on which it is based appear inaccurate, yielding a significant overestimation of lower limb muscle power compared to the gold standard methodology in older adults. Provided that corrected assumptions are used, 5STS field test is confirmed a valid time- and cost-effective field method for the assessing of lower limbs' MP. This approach guarantees a practical and valid alternative to the gold standard methods classically used to measure this important determinant of current and prospective health, mobility, and independent living in aging.

**Author Contributions:** Conceptualization, L.F., G.B., A.S. and S.P.; Data curation, L.F., G.B. and A.B.; Formal analysis, L.F., G.B. and A.S.; Funding acquisition, S.P.; Investigation, L.F., G.B. and A.B.; Methodology, L.F., G.B., A.S. and S.P.; Project administration, S.P.; Software, A.S.; Supervision, F.L. and S.P.; Writing—original draft, L.F.; Writing—review & editing, S.P. All authors have read and agreed to the published version of the manuscript.

**Funding:** This research was funded by Fondazione Cassa di Risparmio di Trento e Rovereto (Caritro), grant number 2020.0409.

**Institutional Review Board Statement:** The study was conducted in accordance with the Declaration of Helsinki, and the protocol was approved by the Ethics Committee of the University of Verona (28/2019).

**Informed Consent Statement:** Written informed consent was obtained from all subjects involved in the study.

**Data Availability Statement:** The data presented in this study are available on request from the corresponding author. The data are not publicly available due to restrictions (privacy).

**Conflicts of Interest:** The authors declare no conflict of interest.

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
