# Peer review of "Estimating Muscle Power of the Lower Limbs through the 5-Sit-to-Stand Test: A Comparison of Field vs. Laboratory Method"

_applsci, doi:10.3390/app122211577_

Round 1
Reviewer 1 Report
The paper deals with a very interesting topic, namely the estimation of muscle strength through sitting and lifting tests.
The paper is well done. The study main purpose is to test the accuracy of the assumptions of Alcazar’s method by using a fully objective, gold standard, and automated method. The main findings showed that the original Time, Position, and Force assumptions of the Alcazar’s method were all overestimated when compared with the laboratory method.
I have no recommendations on improving the work.
Reviewer 2 Report
Please see the attached document

Reviewer 3 Report
1. PICO standards are distorted. The description of the methods in the introduction, and later extended even to the test results, where there is also a duplicate presentation of the data (verbal and tables/figures).
2. Captions under figures are often illegible and describe research methods and tools.
3. In many cases, the lack of precision in determining the significance level of statistical differences and their values in the conclusions.
Reviewer 4 Report
Dear all,
The manuscript fits with the aim of the journal, and the subject reveals good content for researchers and professionals in the sport and exercise medicine field. However, some minor points are listed below:
TITLE
No comments.
ABSTRACT
Line 17: It should be sixty-four and not 64.
Line 24: What is the clinical relevance of this work?
1. Introduction
No comments.
2. Materials and Methods
2.1 Subjects
Under the Materials and Methods section (line; 69): Sixty-four older adults (34 males+ 29 females= 63). Sixty-four or 63?
You mentioned in line 70, participants eligible for this study were at least 60 years old of age, on the other hand, in Table 1, for females, age ranges (min-max) are (57.0-79.0), then you included participants with ages less than 60 y old.
Your measurement methods must be given in detail. Measurement accuracy is necessary to report. Please make sure that your results are given with the same accuracy as the methods. Information about measurement accuracy is important.
Please give information about test-retest reliability measurement if necessary.
Line 73: Please give ID number of the approval with the name of the university or the institution that gave the IRB approval and the.
3. RESULTS
Line 237: in Table 3, please check Males (n = 35), and previously mentioned in line 69 as 34 males.
In Tables 3&4 If your methods allow one or two or three decimals, the result should also be the same decimals through whole results.
4. Discussion
Limitations of your study must be mentioned and discussed in detail somewhere close to the end of the Discussion section. This is always an important part of every manuscript and is something that will lead to new scientific studies in the future.
What is the clinical relevance of this work? Should be added and clarified. At the end, please mention the clinical relevance of your work.
With my best regards,
